# Chronic Facial Pain: Trigeminal Neuralgia, Persistent Idiopathic Facial Pain, and Myofascial Pain Syndrome—An Evidence-Based Narrative Review and Etiological Hypothesis

**DOI:** 10.3390/ijerph17197012

**Published:** 2020-09-25

**Authors:** Robert Gerwin

**Affiliations:** Department of Neurology School of Medicine, Johns Hopkins University, Baltimore, MD 21287, USA; bbgerwin@gmail.com

**Keywords:** chronic facial pain, trigeminal neuralgia, persistent idiopathic facial pain, myofascial pain syndrome

## Abstract

Trigeminal neuralgia (TN), the most common form of severe facial pain, may be confused with an ill-defined persistent idiopathic facial pain (PIFP). Facial pain is reviewed and a detailed discussion of TN and PIFP is presented. A possible cause for PIFP is proposed. (1) Methods: Databases were searched for articles related to facial pain, TN, and PIFP. Relevant articles were selected, and all systematic reviews and meta-analyses were included. (2) Discussion: The lifetime prevalence for TN is approximately 0.3% and for PIFP approximately 0.03%. TN is 15–20 times more common in persons with multiple sclerosis. Most cases of TN are caused by neurovascular compression, but a significant number are secondary to inflammation, tumor or trauma. The cause of PIFP remains unknown. Well-established TN treatment protocols include pharmacotherapy, neurotoxin denervation, peripheral nerve ablation, focused radiation, and microvascular decompression, with high rates of relief and varying degrees of adverse outcomes. No such protocols exist for PIFP. (3) Conclusion: PIFP may be confused with TN, but treatment possibilities differ greatly. Head and neck muscle myofascial pain syndrome is suggested as a possible cause of PIFP, a consideration that could open new approaches to treatment.

## 1. Foreword

Clinical medicine places us between the certainty of diseases described and defined with clear language, with no uncertainty about the criteria for diagnosis, and the reality of everyday ambiguity. Facial pain places us between the realm of our comfort zone of certainty and the sea of ill-defined disorder. When faced with ambiguous complaints, a convenient fallback is to blame psychological distress or anxiety, always placed on the patient, in order to avoid the sense of inadequacy in ourselves. There are certainly clearly defined conditions that cause facial pain. Well-described pain, the affected region well delineated, the causes well known, and our course of action clear, just follow the algorithm. Nevertheless, there are the confounders that disturb our complacency. The pain is in the face, but it is dull, not so well defined, lasting for variable periods of time, altogether too long or even chronic and persistent. We might say that the patient is putting us on, malingering, or is in a life crisis. We try to fit the diagnosis into a known box, or we dismiss the patient, or we pass the patient on to the (next) pain expert. That has been my experience in working with some patients referred to me for management of atypical facial pain or atypical trigeminal neuralgia, with a note from the referring doctor that the patient failed medical treatment for tic douloureux. That experience, repeated many times, prompted this review.

### Introduction

Facial pain, for all its rarity, can be a significant cause of morbidity when present. The two types of non-odontological causes of facial pain that appear to be the most likely to be mistaken one for the other are trigeminal neuralgia (TN) and what used to be called atypical facial pain, but that is now called persistent idiopathic facial pain (PIFP). Confusion between causes of facial pain persists despite the fact that the diagnosis of classical TN should be rather straightforward and not present diagnostic difficulties to the trained clinician [1,2,3,4]. (The term classical TN is generally restricted to TN caused by neurovascular compression.) The caveat is that secondary causes of TN need to be considered, and the cause of classical TN needs to be established for reasons that will be discussed later. A common mistake that should not be made is to treat TN medically without establishing the cause. PIFP, on the other hand, is a diagnostic problem that confronts us head on. Clearly stated guidelines are in fact ambiguous. Descriptive terms include dull, poorly defined, non-localized. My perspective on these conditions is that of a neurologist involved in the diagnosis and management of acute and chronic neurologic and neuromuscular pain for over four decades. That perspective certainly colors my approach to facial pain. I have often been asked to manage patients who had been diagnosed as having TN, but who in fact had PIFP. This review will address that confusion.

This presentation is in the form of a narrative review [5]. There are many recent systematic reviews and meta-analyses of TN so that another is not warranted at this time. At the same time, there is not enough literature available of a type that lends itself to systematic reviews and meta-analyses of PIFP. Furthermore, I have a point of view to present about PIFP that is best expressed in a narrative review. This discussion will address causation, diagnosis and management. Each of these topics is more straightforward for TN than for PIFP, and the accumulated literature for TN is far greater as it has been known for a considerably longer time than PIFP [3]. This review starts with a background discussion of head and facial pain, and then proceeds to an in-depth discussion of TN, PIFP, and of a potential, but mostly overlooked cause of PFIP. The purpose of this review is to clearly differentiate TN from PIFP, to put these diagnoses in perspective with other diagnoses of facial pain, to provide an up to date review of treatment of the two conditions, and to propose a mechanism for PIFP that may be relevant to a subset of patients with that diagnosis.

## 2. Literature Search

This review is based on the selection of systematic reviews and meta-analyses, other review articles, randomized controlled studies, and other studies deemed relevant, published in English (with one exception), for the most part from 2010 to 2020, as found in Medline PubMed, Embase, and in the Cochrane Review. Search terms included trigeminal neuralgia, classical trigeminal neuralgia, secondary trigeminal neuralgia, facial pain, atypical facial pain, persistent idiopathic facial pain, head pain, myofascial pain, orofacial pain, cephalalgia, trigeminal autonomic cephalalgia, post-herpetic facial pain, treatment of trigeminal neuralgia, treatment of persistent idiopathic facial pain, botulinum toxin treatment of trigeminal neuralgia or of facial pain, trigger point dry needling, rhizotomy of trigeminal nerves, radiofrequency treatment of trigeminal neuralgia, gamma knife surgery of the trigeminal nerve, radiosurgery of the trigeminal nerve, microsurgical treatment of trigeminal neuralgia, and microvascular compression. One hundred and one articles were selected for inclusion in this review, of which 18 were systematic reviews and/or meta-analyses.

## 3. Discussion

Facial and non-headache head pain are significant causes of discomfort and disability worldwide, though not as common as tension-type and migraine headaches, which have a global prevalence 38% and 10%, respectively [6]. Included in this discussion of facial pain is pain that affects the eye or below (caudal to the eye) down to and including the jaw/mandible. Pain that occurs in this area may also be accompanied by pain that is above the eye or pain that is frontal, parietal, or occipital. One clinic found that migraine headache occurred in almost 60% of their patients, trigeminal autonomic cephalalgias (TAC) approximately 12%, and of those with facial pain, almost 31% had persistent idiopathic facial pain (PIFP) [4]. These figures reflect the particular pattern of patients referred to that clinic, but illustrate the variation of headache frequency from the global prevalence to individual clinic experiences. The low incidence and prevalence of these conditions do not necessarily reflect the considerable pain and potential disability that accrues from these disorders. The prevalence of TN ranges from 0.03% (95% CI 0.01–0.08) to 0.3% (CI 0.16–0.55), with women three times more likely to be affected than men [7]. Migraine and tension-type headache will not be discussed further in this report, as they are not primary causes of facial pain. Odontologic disorders are a known cause of facial pain but will not be addressed in this review except in passing as temporomandibular joint disorders are mentioned as a cause of myofascial dysfunction in relation to PIFP.

There are four major types of non-odontogenic head or facial pain. They are (1) trigeminal autonomic cephalalgias (TAC), (2) trigeminal neuralgia (TN), (3) post-herpetic neuralgia (PHN) and (4) persistent idiopathic facial pain (PFIP). Myofascial pain of head and neck muscles will be discussed as a potential explanation for a subset of PIFP, as well as an exacerbating factor for TN.

The conditions that cause facial pain other than TN may not be well known to many physicians, as they are relatively uncommon compared, for example, with tension-type and migraine headache.

### 3.1. Trigeminal Autonomic Cephalalgias

The TACs are a group of disorders that result in pain within the trigeminal nerve distribution. The most common of these is cluster headache. The others, paroxysmal hemicrania, short-lasting unilateral neuralgiform headache attacks with conjunctival injection and tearing (SUNCT) and short-lasting unilateral neuralgiform headache attacks with cranial autonomic symptoms (SUNA), are much less common. The prevalence of paroxysmal hemicrania is estimated to be 0.5/1000 or less. That of hemicrania continua is so infrequent that its prevalence is not determined. SUNCT and SUNA have a prevalence of 0.5 to 1.0 per 1000 [8]. The diagnosis of these conditions is usually made based on the history and the exclusion of other diagnoses. It is important to tailor the history so as to elicit the symptoms that lead to the proper diagnosis.

#### Cluster Headache

Cluster headache is the most common and therefore perhaps the most familiar of TACs. The prevalence of cluster headache in adults of working age in Sweden in 2010 was 0.054% [9]. As in TN, disability associated with cluster headache is significant, with almost twice as many cluster headache individuals out on sick leave than a control population (17.30% compared to 9.16%) [9]. Women are more likely to be on sick leave than men (25.31% vs. 13.38%) and more likely to be receiving a disability pension than men (13.17% vs. 8.79%). A 2008 meta-analysis of studies with widely varied lifetime prevalence showed a pooled prevalence of 0.12% [10]. The male:female ratio was 4:3, but the ratio seems to be changing, with more women being diagnosed than men [8]. The lifetime prevalence in three European countries (Sweden, Germany, Italy) ranged from 0.15% to 0.28% [11]. Cluster headache presents with sudden onset of unilateral head pain, usually involving the eye and hemicranium, accompanied by parasympathetic autonomic manifestations such as tearing, nasal congestion, forehead sweating, miosis and ptosis. The autonomic manifestations distinguish it from first or ophthalmic trigeminal division TN. Moreover, the temporal pattern with prominent night attacks and clusters of attacks interspersed with periods free of attacks distinguishes it from TN. The duration of an attack, generally lasting from 15 to 30 min, is considerably longer than a single, lightning attack of TN. Furthermore, absence of typical sensorimotor triggers in cluster headache distinguish it from TN.

### 3.2. Trigeminal Neuralgia

Trigeminal neuralgia is more common than TAC, and therefore more likely to be encountered by internists, family practice physicians and general practitioners. TN, however, may be confused with other facial pain, particularly PIFP, which is thought to have both neuropathic and non-neuropathic causes. The prevalence of TN around the world varies from 76.8 per 100,000 or 0.0768% to 29.5 per 100,000 persons or 0.0295% [12,13,14]. The incidence of TN, as reported in one systematic review, ranged from 12.6 to 28.9 per 100,000 person years [15]. However, TN incidence rates vary with comorbid conditions, as reflected in the increased incidence of TN in migraineurs (incidence rate of 136.39 versus 20.06 per 100,000 person years, respectively, for migraineurs and for matched controls) [16]. Trigeminal neuralgia is as much as 15–20 times more common in persons with multiple sclerosis than in the general population [17,18,19,20]. Persistent idiopathic facial pain, on the other hand, is relatively rare. The lifetime prevalence for TN and for PIFP in Germany was estimated to be 0.3% (95% CI 0.1–5%) for TN versus 0.03% (95% CI < 0.08%) for PIFP [21].

Trigeminal neuralgia, or tic douloureux, is chronic, episodic, and recurrent. It is a neuralgic pain that at best is distracting and at worst totally debilitating and disabling [22,23]. Pain originates in the trigeminal nerve complex or its territory. It is caused by compression of the nerve root by vessels in the posterior fossa in 80–90% of cases [22,23,24], but it can be secondary to a number of other conditions to be mentioned shortly. The vessel most commonly compressing the trigeminal nerve root is the superior cerebellar artery (SCA), as identified by high-resolution 3 T magnetic resonance imaging (MRI). The anterior inferior cerebellar artery (AICA), the basilar artery, and an ectatic vertebro-basilar artery have also been implicated in some patients. The trigeminal nerve can also be compressed by veins, particularly the superior petrosal and transverse pontine veins [25]. High-resolution MRI is highly effective in identifying neurovascular compression of the trigeminal nerve as well as to identify lesions that cause secondary TN that occurs in approximately 15% of cases [26]. Trigeminal neuralgia can also be caused by lesions elsewhere along the nerve as in the root entry zone in multiple sclerosis (MS) [27]. There is frequently a concomitant neurovascular compression of the trigeminal nerve in MS in addition to a pontine demyelinating trigeminal root entry zone plaque, causing a ‘double-crush’ lesion [28]. Trigeminal neuralgia can also result from tumor and arteriovenous malformation compression of the nerve or the gasserian (trigeminal) ganglion. Deformation of the nerve from adhesions following microvascular decompression has also been reported to cause a relapse of TN [29]. Likewise, inflammation and trauma can lead to TN. Minute tumor-adjacent vasculature thought to contribute to clinically-symptomatic TN, not seen on conventional low-field imaging, has been identified by seven-tesla ultra-high-field multimodal magnetic resonance imaging using co-registering time-of-flight angiography and T2-weighted turbo-spin echo images [30]. TN can also be caused by solitary pontine lesions [31]. Patients with solitary pontine lesions failed to respond to surgical microvascular decompression treatment. An estimated 2% of cases are hereditary, most often associated with Charcot–Marie–Tooth disease [32,33,34]. Trigeminal neuralgia is characterized by very short bursts of unilateral, lightning, electric-like sensations (Table 1) [35]. There is a brief refractory period following a burst of pain.

The pain is confined primarily to the distribution of just one of the three divisions of the trigeminal nerve. The most commonly affected division is the third, mandibular, division; next is the second, maxillary, division, and least often affected is the first or ophthalmic division. However, when the pain is located near the boundary of another division of the nerve, it may seem to spread to that neighboring division [36]. Though often spontaneous, TN frequently has clear triggers such as touching a sensitive or trigger spot, brushing teeth, speaking, chewing, eating or drinking hot or cold food or liquids, with approximately 2/3 of patients having just one trigger spot or stimulus. All such triggers, however, are generally innocuous stimuli in normal individuals. Patients can usually point with one finger to the trigger zone, and can report the activities that trigger an attack. Most individuals suffering from trigeminal neuralgia have both spontaneous and stimulus evoked pain episodes; that is, pain that is not triggered by an identifiable stimulus, and pain that is. The pain episodes are brief, lasting seconds, with a lingering post-shock pain that lasts a minute or two, but that lacks the sharp, penetrating pain of the initial shock. In mild cases, there may be only a few attacks per day, but in severe cases, there may be recurrent trains of attacks that last minutes to hours. The affected individual may be immobilized by fear of initiating an attack. The attacks may continue for days, weeks, or months until they remit, decreasing in frequency or stopping altogether. They invariably return, relapses recurring more frequently with age.

Devor and his colleagues have studied the nature of the ‘shooting pain’ and its neural substrates [37]. Background pain was found in 39% of their patients. Pain was rated as the worst pain imaginable in 50% of their patients. Paroxysms of pain occurred from 10 to 100 times per day in 47.3% and hundreds of times per day on typical (not ‘bad’) days, in 7.3%. Spread of pain in TN is much less common than other radiating pains, such as sciatic radiculopathy. Half of the TN patients reported no spread. In the majority of patients reporting spread or expansion of their pain, the spread remained within the bounds of one division of the trigeminal nerve. The authors attributed the paroxysmal pain to hypersensitivity in the topographically arranged trigeminal ganglion and trigeminal nerve root, where a traveling wave could lead to pain in one or an adjacent division of the trigeminal nerve, but would not lead to pain in the posterior scalp, neck or shoulder as they are not represented in the trigeminal ganglion or nerve root.

Individuals in whom attacks of pain last minutes to hours, or are persistent or chronic, waxing and waning over the course of the day, or in whom pain extends beyond one division of the trigeminal nerve, may still be mistakenly diagnosed as having trigeminal neuralgia. Such individuals may point to one side of the face as the site of their pain or may indicate that pain is bilateral. Their pain may be further atypical in lacking the usual triggers of pain such as brushing teeth or touching a trigger area. Such pain that is atypical for TN is a different kind of facial pain than classical TN. However, even in cases that are not characteristic trigeminal neuralgia, chewing, and even speaking, for example, may be triggers. Chewing and speaking activate orofacial and neck muscles, and are accompanied by small movements at the cervical–cranial junction. Nociceptive sites in these muscles may be activated by chewing or speaking. Patients with atypical facial pain are unlikely to have trigeminal neuralgia, and more likely to have what is now called persistent idiopathic facial pain (PIFP). In truth, many patients seen in my clinic, referred for evaluation of atypical facial pain, had a mistaken diagnosis of trigeminal neuralgia. Many had, instead, orofacial myofascial pain syndrome, with pain coming from trigger points in the facial or neck muscles, including the muscles that control mouth opening and closure (temporomandibular joint-related muscles) or that control lip motion and therefore such actions as smiling, chewing and talking. In such cases, neck and shoulder muscles were usually involved, primarily because mouth opening and closing is associated with changes in the spatial relationship between the head and the upper cervical spine, moving the head backwards or forwards on the neck.

In summary, there are cases of facial pain that are truly trigeminal neuralgia (primary or secondary), cases of PIFP alone, without trigeminal neuralgia, and there is a third group of patients that we have seen that have both trigeminal neuralgia and also PIFP but with myofascial trigger points in head and neck muscles. This classification is purely anecdotal, as there are no studies of the relationship between TN, PIFP, and myofascial pain in the literature. Nevertheless, we found that in patients with both trigeminal neuralgia and orofacial muscle trigger points, the TN aggravated the trigger points and activation of the trigger points served as a trigger for attacks of trigeminal neuralgia. Successful management of both conditions simultaneously was required in order to suppress both the episodic pain of TN and the persistent pain of orofacial muscle pain.

#### Diagnosis of Trigeminal Neuralgia

The diagnosis of classical TN is made on the basis of a characteristic history of lightning-like sharp, electrical pain that is felt in one division of the trigeminal nerve, leaving a dull after pain that lasts for a variable, usually short, period of time. There is often a trigger, but there does not need to be one. The attacks are typically infrequent at first, but become more frequent with the passage of time, and may increase in frequency to occur hundreds of times a day. Remissions occur, but relapses become more frequent with aging. There is no dullness or loss of feeling reported. Some patients tell atypical stories in which pain crosses divisions of the trigeminal nerve, or paroxysms of pain last longer than lightning attacks of pain. The neurological examination is normal in classical TN. Motor and sensory examination of the face in particular is normal in classical TN, but is useful in identifying secondary trigeminal nerve dysfunction that could lead to a diagnosis of secondary TN or trigeminal neuropathy. The same is true of the blink and other trigeminal reflex tests, as the presence or absence of an abnormal result does not affect the diagnosis of TN, but may indicate a need to examine for causes of secondary TN.

Every case of trigeminal neuralgia should be investigated with an MRI scan in order to evaluate the pontine region for neurovascular compression, since that can be treated with microvascular decompression. There is a small but important percentage of patients who will have a different kind of structural lesion compressing the trigeminal nerve. There is no data available on the incidence of first identification of multiple sclerosis (MS) by the diagnosis of TN, but it is certain that MS-related TN occurs in patients with an established diagnosis of MS that can be confirmed by MRI imaging [17,18,19,20,27,28].

Electrodiagnostic testing may be useful in distinguishing primary from secondary trigeminal neuralgia. The European Academy of Neurology Guidelines on trigeminal neuralgia [38] states that trigeminal reflex testing has high specificity and sensitivity for secondary trigeminal neuralgia, whereas evoked response testing has high sensitivity but low specificity and does not reliably distinguish between primary and secondary TN.

### 3.3. Management of Trigeminal Neuralgia

The goal of treatment of trigeminal neuralgia is to reduce the frequency and intensity of the paroxysms of facial pain or eliminate them altogether. Most management studies used pain relief as the major outcome measure, though there is no consensus about outcome measures [39]. Pain intensity is not the same as the degree of pain relief. Most studies of treatment of TN used either the visual analogue scale (VAS) for pain or the Barrow Neurological Institute Pain Intensity Scale to measure pain intensity. A smaller number used attack frequency, measures of daily functioning, emotional impact of TN, or satisfaction with treatment [39].

#### 3.3.1. Pharmacologic Treatment

Initial treatment is almost always pharmacologic, the goal being to decrease hyperexcitable bursts of nerve discharges. Additional therapeutic approaches include interventions such as percutaneous rhizotomy, radiofrequency thermocoagulation, balloon compression, botulinum toxin, gamma knife radiosurgery, and microvascular decompression. Today, there is a rather large number of potential pharmacologic and interventional therapies for TN [40]. Even so, initial treatment of TN is usually pharmacologic, even for secondary TN, utilizing anticonvulsant medications. Carbamazepine is generally considered the drug of choice. It is effective in 60–100% of cases, at least for a time, though the failure rate of long-term treatment can be as high as 50% [26,38]. Furthermore, there are significant adverse side effects associated with carbamazepine. Oxcarbazepine, considered to have fewer side effects than carbamazepine, is often used instead of carbamazepine, although there is little experimental data as opposed to clinical experience to support its use. These drugs target voltage-gated sodium channels. There is low quality evidence for the use of other anticonvulsant drugs such as lamotrigine and gabapentin [38]. Newer pharmaceuticals such as eslicarbazepine, an active metabolite of oxcarbazepine, and an Nav1.7 channel blocker vixotrigine, are being evaluated as treatments [40].

#### 3.3.2. Botulinum Toxin Type A (BTxA)

Botulinum toxin type A has a beneficial role in the treatment of neuropathic pain [41,42,43,44,45,46]. It has both antinociceptive and anti-inflammatory activity, the two mechanisms being dissociated. Botulinum toxin A acts at both peripheral and central sites. Peripherally, it blocks the docking of intraneuronal vesicles to the inner membrane of the nerve terminal inhibiting the release of neuropeptides and neurotransmitters. Consequently, the extracellular concentrations of acetylcholine, substance P, serotonin, calcitonin gene-related peptide (CGRP), glutamate, and proinflammatory mediators are decreased. Plasma CGRP levels decrease in TN patients who respond well to BTxA treatment, whereas non-responders show no decrease in plasma CGRP levels [41]. Centrally, botulinum toxin A acts at the spinal dorsal horn as a result of retrograde toxin transport. Microglial activation, an important component of nociception, is also attenuated [43,47]. Furthermore, BTxA inhibits sodium ion channel activity. A single case report of TN treatment by BTxA was published in 2002 [48]; since then have been a small number of studies and generally they included a relatively small number of subjects. One randomized controlled trial of 40 subjects in whom structural lesions were excluded showed significant benefit [49]. BTxA was administered in the area of pain by means of both subcutaneous and submucosal injections. The major adverse effect was transient facial weakness. Two systematic reviews and meta-analyses of the efficacy and safety of BTxA treatment of TN were published in 2016 [50,51], citing the same four randomized controlled trials with a total of 178 patients. Ninety-nine patients received BTxA and 79 received placebo treatment. There was no standardized dosage or method of injection, the doses of onabotulinumtoxin A ranging from 25 to 100 units. Injections were generally administered either subcutaneously or intradermally in the region of clinically evident pain. The intensity of pain and the frequency of attacks were both significantly lower with BTxA compared to placebo, the benefit lasting 3 months. Transient facial asymmetry and edema were the two main adverse effects and were said to be well tolerated. A non-randomized, uncontrolled, unblinded study of 27 subjects evaluated the effect of BTxA over 6 months. BTxA was injected about the mandibular branch of the trigeminal nerve, around the pterygopalatine ganglion and the maxillary branch of the trigeminal nerve near the trigeminal ganglion. A total of 63% of subjects had a greater than 50% reduction in pain after the first week, 74.1% achieved that after the second month, and 88.9% at the end of 6 months. 15/27 subjects required a second injection approximately 2 months after the initial injection, and 7/15 required a third injection at a mean of 87 days. 44% were pain-free at 6 months. There was a similar decrease in frequency of attacks per day from a baseline of 217.7 +/− 331.5 to 55.15 +/− 196.3 at the sixth month. Adverse effects were infrequent, with one patient experiencing facial weakness that cleared by the second month, and two patients experiencing ipsilateral masseter muscle weakness that was permanent but mild [52]. An open-label study of the effect of BTxA injection of the sphenopalatine ganglion in 10 subjects with maxillary (second division) TN showed a decrease in pain at 4 weeks from a VAS score of 8.1 +/− 1.0 to 1.9 +/− 1.4 and a decrease in the daily attack frequency from 19.4 +/− 8.8 to 4.9 +/− 5.4 [53]. Another pilot study of BTxA injected in the sphenopalatine ganglion region in 10 subjects showed similar results [54]. There were four responders with at least a 50% reduction in attack frequency among the nine subjects completing the trial. Pain intensity in the responders decreased from a mean of 5.8 +/− 2.1 to 3.65 +/− 3. Adverse effects were mild and transient, although one patient had diplopia that lasted 1 month. The authors reported that all subjects had involvement of the maxillary division of the trigeminal nerve, that 9 of the original 10 subjects also had involvement of the mandibular division, and 7/10 also had involvement of the ophthalmic division. This is unusual, as TN most commonly affects primarily just one division, raising the issue that the patients may have been misdiagnosed. It is possible that the difference in presentation from classic TN accounted for the relatively few subjects who were responders [54].

We can conclude that BTxA offers an effective form of treatment for those individuals for whom oral medication such as oxcarbazepine has failed or for whom interventional therapies such as peripheral nerve ablations or microvascular decompression are not suitable.

#### 3.3.3. Interventional Approaches

##### Percutaneous

The percutaneous therapeutic approaches are ablative procedures that are directed to the trigeminal (gassarian or semilunar) ganglion located in Meckel’s cave [55,56]. The three common ablative techniques are chemical (glycerol rhizotomy), mechanical (balloon compression) and thermal (radiofrequency thermocoagulation). The goal in treatment is to selectively destroy the A delta and unmyelinated C fibers that mediate pain, while preserving the A alpha and beta fibers that mediate touch [57]. In general, these techniques have an acceptably high rate of initial pain and attack reduction, but the benefit lessens over time. The procedures can be repeated if necessary. A significant drawback is the potential for the development of anesthesia dolorosa. Other adverse effects include weakness of facial and masticatory muscles and facial anesthesia. The percutaneous procedures may be used when pharmacologic therapy such as carbamazepine or a similar drug does not adequately control the pain.

There is one systematic review and meta-analysis of the comparative safety and efficacy of per-cutaneous approaches for the treatment of trigeminal neuralgia, comparing each of the three types of treatment with each other [58]. Radiofrequency thermocoagulation had significantly greater odds (OR, 2.65; 95% CI: 1.29–5.44; I2: 85.5%) for immediate pain relief than glycerol rhizotomy. The rates of pain recurrence over 5 to 30 months were similar between the two groups. There was a significantly higher risk of anesthesia with radiofrequency rhizotomy. The rates of complications such as anesthesia dolorosa, keratitis, and weakness of chewing were the same in the two groups. The rates of immediate pain relief and of pain recurrence over the long term (6–28.5 months) were similar for balloon compression and glycerol rhizotomy. The risk of mastication weakness was significantly (9-fold) higher for balloon compression compared to glycerol rhizotomy and diplopia was likewise more likely with balloon compression (4.8% compared to 0.45% for glycerol rhizotomy). However, mastication weakness was usually not clinically significant, and remitted over the course of several weeks. Rates of immediate pain relief, of pain recurrence, and of adverse effects were similar for balloon compression and radiofrequency thermocoagulation. Post-operative herpes eruption was more likely with radiofrequency thermocoagulation than with glycerol rhizotomy. As of 2019, there were no randomized trials comparing all three techniques together. Moreover, the studies were not blinded or randomized. Additionally, procedure techniques change over time, and vary from study to study, limiting the reliability of meta-analyses of studies conducted over many years. For example, there is no agreement about the optimal temperature to be used for radiofrequency thermocoagulation. Radiofrequency can be pulsed or continuous, and at different temperatures [59]. Computerized tomography guidance has been proposed to decrease procedure time, anesthesia time, exposure to radiation and to improve the accuracy of needle placement [60]. Perhaps the most feared complication of the percutaneous procedures is anesthesia dolorosa, a severe and difficult to treat persistent pain that occurs rarely, up to 1% of cases [61], though it is usually much less frequent. The complication rate associated with radiofrequency thermocoagulation is directly related to the temperature used. Effective treatment associated with the lowest rate of adverse effects was achieved at temperatures of 65–70 degrees centigrade [62].

Variations of the peripheral approach to nerve ablation include endoscopic peripheral nerve ablation [63], cryotherapy nerve ablation [64], and acupuncture [65].

##### Gamma Knife

Gamma knife stereotactic radiosurgery (GKRS) is a minimally invasive approach for the management of trigeminal neuralgia refractory to medication [66,67]. Gamma knife radiosurgery (GKRS) produces axonal degeneration, ion channel destruction, and an electrophysiologic block that reduces nociceptive input [56]. Highly precise targeting of the trigeminal nerve or the trigeminal ganglion is possible, limiting adverse effects. The preferred target for GKRS has evolved from the trigeminal ganglion itself to the retrogasserian region and to the root entry zone. A meta-analysis of GKRS outcomes reported that pain relief ranged from 69 to 85% at one year, falling to 38 to 52% at 5 years and 30 to 45.3% at 10 years [56,68]. Onset of pain relief is delayed, ranging from 15 to 78 days on average, up to 6 months. As is the case with percutaneous treatments, retreatment is possible, but, though effective, there is a greater risk of complications [69]. The benefit of repeated GKRS has been greater in those subjects who also had facial sensory loss.

A number of systematic reviews and meta-analyses compared GKRS with microvascular de-compression (MVD), which will be discussed below in connection with MVD, the only treatment that directly corrects the cause of TN in at least 50% of cases [70,71,72,73,74,75,76].

##### Microvascular Decompression

Microvascular decompression (MVD) is the gold standard of treatment for TN caused by vascular compression of the trigeminal nerve [71,77]. A meta-analysis of 46 studies totaling 3897 patients showed long-term freedom from pain in 76% of patients [70]. A greater likelihood of a successful outcome was associated with a duration of 5 years or less, compression by the superior cerebellar artery (SCA), compression by an artery, including the anterior inferior cerebellar artery (AICA), rather than a vein, and classical (rather than atypical) TN. Patients with classical TN were more likely to have arterial compression and patients with more persistent or atypical pain were more likely to have venous compression. Adverse effects were relatively few compared to percutaneous ablation techniques or GKRS. Adverse effects included facial numbness (5.5 to 13.9%), facial dysesthesia (5.3–5.7%) minor hearing impairment (2.7%), and dizziness (1.8%). More serious complications such as cerebrospinal fluid leaks and infection are rather rare, and mortality is quite uncommon (0.0–0.4%).

A small subgroup of patients have TN caused by vertebro-basilar artery ectasia. The patients in this subgroup tend to be older than in TN with SCA or AICA involvement. The outcome after open MVD in this subgroup was comparable to the outcomes in open MVD of the more common smaller artery compression of the trigeminal nerve, but the complication and recurrence rates were lower [78].

The two main approaches to MVD are open microscopic microvascular decompression (OMVD) and minimally invasive endoscopic microvascular decompression (EMVD). OMVD was pioneered by Janetta [79], but has a number of problems associated with it that are minimized with the EMVD. Specifically, visualization is improved and instrument breakage with consequent neural and vascular injury is less with the EMVD. Post-operative complications such as hearing loss, spinal fluid leakage, and facial paresis or paralysis are significantly less with EMVD [80]. The first comparative meta-analysis of the two approaches was published in 2017 [81]. Thirteen studies of OMVD and 10 studies of EMVD, with a total of 6749 patients, were included in the meta-analysis. Of these, 993 had undergone EMVD. Good pain relief was achieved in 81% of OMVD procedures, and in 88% of the EMVD patients. The mean recurrence rate was 14% in the former and 9% in the latter. The rate of complications (facial weakness, hearing loss, cerebellar injury, infection, and death) was 19% for OMVD and 8% for EMVD. Improvements in surgical techniques in more recent years may bias these figures, as could the asymmetry in the numbers of patients in each treatment group, and the great heterogeneity among the studies. Another meta-analysis of nine studies that evaluated MVD in a number of cranial nerve syndromes caused by nerve compression, including hemifacial spasm and glossopharyngeal neuralgia, in addition to TN. As in the meta-analyses of MVD for treatment of TN, this study also showed a significantly better outcome for EMVD than OMVD. The remission rate for EMVD was 1.71 (OR, 1.71, 95% CI 1.14–2.54; *p* = 0.0089) [80]. Perioperative complications were 1/3 less for EMVD compared to OMVD. 

Microvascular decompression had better outcomes (greater freedom from pain) with lower rates of facial numbness, but higher rates of perioperative complications, than RF thermocoagulation [71]. However, a problem with the comparison is that the etiology of TN could have been different. Those selected for MVD had neural compression by a vessel and those selected for RF either may not have any vascular compression of the trigeminal nerve or could not undergo MVD. Furthermore, the MVD series included both endovascular and microscopic procedures. Thus, rates of remission may be looking at two different populations, whereas the complication rates may necessarily have to be parsed between perioperative complications versus recurrence rates.

MVD has been compared to one or more trigeminal nerve ablation techniques in both single-institution studies and in meta-analyses. Pain and post-operative complications 5 years after treatment with either MVD or partial sensory rhizotomy (PSR) was evaluated by a self-completion assessment set of questionnaires. The quality of life as assessed by the SF-12 was lower for patients in both treatment groups than in healthy controls, whereas significant anxiety was greater in the PSR group. Patients in the PSR treatment group had more post-operative complications and of greater severity than those treated by MVD, including pain, numbness, dysesthesias, and difficulty eating [82]. MVD was compared to GKRS in a meta-analysis that looked at 414 studies of GKRS, 12 MVD and 5 of both types of treatments, published between 2005 and 2015 [76]. The type of MVD was not indicated in this study, and both EMVD and OMVD were included. As always, the cause of TN was likely to be generally different in the two groups, at least in the younger population that was more likely to undergo MVD, because GKRS is most likely to be used in patients without vascular compression of the nerve and more likely to be used in an older population where MVD may be considered to be a greater risk. With that caveat in mind, the initial success rate was greater for MSD than for GKRS, 86.9% vs. 71.1%, respectively, the 2 year success rate was 91.4% vs. 77.8%, and after 5 years, 84% vs. 63.8%, respectively, the differences being statistically significant. Vertigo, low in both procedures, was more common after MVD, but anesthesia dolorosa was more common after GKRS. Cranial nerve palsies were few after either procedure, but more common after GKRS. Hearing loss was uncommon in both procedures (under 2%), but dysesthesia was significantly more common in GKRS, 2.3% after MVD vs. 2.8% after GKRS. The average length of recurrence-free intervals was the same for the two procedures—30.40 months with MVD and 30.55 months with GKRS. A more recent systematic review confirmed the previous findings that MVD results in a higher percentage of favorable results with a lower incidence of complications [73]. Another meta-analysis published in 2018 showed an even greater initial success rate of 96% (95% CI 93.3–98.6%) for MVD compared to 71.8% for GKRS [75]. The disparity in outcome with greater pain relief following MVD continued for long-term follow-up as well as for short-term effects [74]. As expected, retreatment rates are higher for GKRS than for MVD, 17% vs. 10.3%, though the difference is not statistically significant (OR, 1.76; 95% CI, 0.93–3.55; *p* = 0.08). However, a systematic review and meta-analysis found that once GKRS had failed, there was no difference in the outcome between repeat GKRS and MVD [72]. One possible factor contributing to this is that the initial choice in selecting GKRS could have been based on the lack of neurovascular compression. Nevertheless, some reports indicated that high rates of initial success with MVD were achieved even when microvascular compression of the nerve was not found at surgery. The advantage of GKRS in having virtually no operative adverse effects and same day discharge home is balanced against the better short- and long-term outcome, fewer complications and a lower rate of repeat operative treatment associated with MVD. Further, as might be expected, patients who do not do well with MVD are generally those without a well-defined vascular compression of the trigeminal nerve, or with only venous compression.

In summary, there is effective treatment for trigeminal neuralgia but many treatments lose benefit over time. For some therapies, such as carbamazepine, the rate of adverse effects is high. For others, such as peripheral nerve ablations, the rate of adverse effects is lower, but can include the very serious complication of anesthesia dolorosa. Even the most effective therapies lose effectiveness over time. (Table 2). New therapies continue to be proposed and tested, as for example peripheral nerve stimulation. However, studies to date of this technique are few and with very small numbers of subjects. 

### 3.4. Persistent Idiopathic Facial Pain (PIFP)

PFIP is rather ill defined, being nothing so much as a dull, poorly localized, facial pain of longer duration as opposed to classical TN. Classical TN is a very well-defined type of pain, rather precisely located within the distribution of a single division of the trigeminal nerve, and well defined temporally. In contrast, PIFP, though of necessity within the distribution of the trigeminal nerve, is not confined to the distribution of a particular trigeminal division. It is further defined by its duration of >2 h per day, and of 3 months or longer duration, as well as lack of any other cause [1] (Table 3).

#### 3.4.1. Diagnosis of PIFP

Description: Persistent facial and/or oral pain, with varying presentations but recurring daily for more than 2 h per day over more than 3 months in the absence of clinical neurological deficits.

The cause(s) of PIFP is (are) elusive. Neurovascular contact with displacement of the trigeminal nerve is associated with TN, but not with PIFP [83]. Quantitative sensory testing of PIFP and of traumatic TN compared to healthy controls showed no significant differences among the groups for warm or cold thermal stimulation, or for tactile stimulation. However, lower detection of mechanical stimulation (pinprick), and subjective numbness and dysesthesia was found in 36% of PIFP subjects. The authors concluded that there was a neuropathic component to PIFP [84]. Neuropathic characteristics were identified in 10% of PIFP subjects using a Douleur Neuropathique 4 (DN4) questionnaire, though the PIFP group was only compared to a cohort of post-herpetic pain patients and not to normal controls [85]. Notwithstanding, the data that suggests that at least some cases of PIFP are neuropathic in nature remains sparse. Estimates of PIFP prevalence range from 0.03 to 1.0% [1].

#### 3.4.2. Differential Diagnosis between PIFP and TN

Despite the sharp contrasts between the two conditions, cases of PIFP continue to be misconstrued as TN, or referred to as ‘atypical trigeminal neuralgia’ [86,87]. This is especially true of patients treated pharmacologically by physicians not well acquainted with either condition, but aware that TN causes facial pain. The neurologic examination is normal in both conditions (in contrast to secondary TN where there are likely to be neurological deficits). The distinction is important to make, because there are treatments effective for TN, but not for PIFP [86]. Nevertheless, there are no positive findings that confirm the diagnosis of PIFP, and the diagnosis remains one of exclusion. If a neuropathic cause is found, the facial pain is no longer idiopathic, but should be termed trigeminal neuropathy of such and such origin, such as traumatic or post-viral.

#### 3.4.3. Differential Diagnosis of Facial Pain

Differential diagnosis of facial pain, orofacial pain, and headache, is important [88], but in fact, there is little overlap between many of the syndromes described and TN or PIFP. Many of the other conditions that affect the head and neck, such as glossopharyngeal neuralgia, migraine and cervicogenic headache, trigeminal autonomic cephalalgias, burning mouth syndrome, and giant cell arteritis, for example, can be distinguished from TN and PIFP by a history that describes the temporal pattern of pain, the location of the pain, and the presence or absence of autonomic phenomena. Physical examination reveals the presence or absence of neurologic deficits. Trigeminal nerve disorders such as trauma, inflammation, or tumor, should be considered. Musculoskeletal elements and odontoid structures must be considered for orofacial disorders that result in facial pain. Laboratory testing, including cranial imaging, is necessary in specific cases. Psychosocial disorders are present in somewhat less than half of the patients with PIFP and burning mouth syndrome together [87,89], but such conditions also commonly exacerbate musculoskeletal conditions. Many patients with PIFP have neuro-sensory disorders, but I would interpret that as indicating that the diagnosis in those cases should be a trigeminal neuropathy, and not PIFP. For example, trauma to the alveolar nerve as a result of tooth extraction can result in severe, persistent, pain, but in my experience is more localized to the mouth and jaw and less to the face itself, though there may also be comorbid myofascial involvement of facial muscles. The authors also say that the clinical presentation of PIFP “is often difficult to distinguish from a chronic myofascial pain…” [88]. It is this last point that seems to be worthy of further investigation. These authors [88] go on to say that regional myofascial pain from pericranial, masticatory and cervical muscles may give rise to pain that clearly overlaps with the pain of PIFP. However, they say that the referred pain patterns from these muscles may refer pain to areas that should not be seen with PIFP, such as the eye, jaw or teeth, implying that if there is pain in those areas, some other cause should be considered than myofascial pain. Others make a similar point, and go on to say that “a large number of PIFP patients present with a history of moderate trauma and subclinical sensory changes… Thus, PIFP is considered a neuropathic pain syndrome” [89]. I would argue, however, that in the cases of trauma with subclinical sensory changes, the diagnosis should be traumatic trigeminal neuralgia or neuropathy, and not PIFP. While it may seem trivial to make this point, the danger is that in considering PIFP patients as having neuropathic pain, non-neurologic causes, particularly myofascial pain syndrome (MPS), may be either overlooked or dismissed. It is important not to overlook MPS because it can be effectively treated.

Migraine headache, cluster headache, as well as the trigeminal autonomic cephalalgias, can also have facial pain complaints [4], but the timing and the associated tearing or conjunctival injection, for example, separate them from TN and from PIFP. However, non-odontoid orofascial muscle syndromes associated with temporomandibular joint dysfunction may give rise to typical PIFP [86].

#### 3.4.4. Treatment of PFIP

Present treatment of PIFP is non-specific and symptomatic, following the guidelines for management of chronic pain [1]. Though opiate therapy is mentioned as a possible approach, caution must be exercised because addiction and misuse are major concerns in a chronic condition of unknown etiology. Tricyclic antidepressant medication such as amitriptyline is a mainstay of PIFP treatment. Some of the selective serotonin reuptake inhibitor antidepressants and some of the anti-epileptic drugs such as the gabapentoids are used in treatment. Biopsychosocial therapy and cannabinoids are also used. Sphenopalatine ganglion blockade gives moderately good relief in refractory cases [86]. Unfortunately, evidence for the effectiveness of treatment for PIFP, whether it be tricyclic antidepressants, cognitive behavior therapy, antidepressant drugs, anti-epileptic drugs, low-level laser, sphenopalatine ganglion block, or treatment of oral mal-occlusion, is all inadequate to make a recommendation (Table 4). Therefore, multimodality approaches are often used in an attempt to provide pain relief and to improve the quality of life.

### 3.5. Myofascial Pain as a Potential Etiology of PIFP

An outstanding issue remains: what are the causes of PIFP? As particular causes are identified, such as trauma to the trigeminal nerve, the pool of PIFP patients decreases. Pain referred to the face from another site gets little mention in the literature of PIFP. The known referred pain patterns of myofascial trigger points in the neck and face muscles strongly suggests that at least a subset of PIFP is caused by myofascial trigger points [90]. Referred pain from the sternocleidomastoid muscle very clearly resembles the pain of PIFP, but other muscles also refer pain to the side of the face (Figure 1).

Specifically, trigger points (TrPs) in the upper trapezius and medial pterygoid and anterior and middle temporalis muscles refer pain to the region of the third division of the trigeminal nerve. Those in the deep masseter muscle refer pain to the region of the second and third divisions. TrPs in the sternocleidomastoid muscle refer pain to the first and second divisions of the trigeminal nerve. Thus, TrPs in one or a combination of several head and neck muscles can result in facial pain that is localized to one or more divisions of the trigeminal nerve. A number of these muscles are related to the function of the temporomandibular joint, or to the functions of mastication and speaking. This is of particular interest since a recent observational study reported that placing the muscles of mastication into their physiologic rest position with minimal tonus of these muscles using an oral orthotic device resulted in a decrease in pain (VAS pre-orthotic: 9.05, SD 0.8 compared to 5.87 ± 0.7 post-orthotic; no statistics for significance were presented), suggesting that pain from the orofacial muscles associated with jaw movement can contribute to PIFP pain [91].

#### Myofascial Pain Syndrome Definition and Diagnosis

Myofascial pain syndrome (MPS) is a collection of symptoms and signs that occur as a result of the presence of trigger points (TrP) in skeletal muscle. The TrP is thought to be a small focus of exquisite tenderness in muscle that when stimulated or irritated refers pain to distant sites in the body. The TrP is considered to be the result of acute or chronic muscle overload that results in a segmental sarcomere contraction and the production of nociceptive neurotransmitters and cytokines [92]. TrPs occur both as comorbidities associated with other conditions such as migraine or tension type headache, and as a primary disorder. The syndrome is diagnosed by history and physical examination by the presence of localized tenderness in a taut muscular band, that when stimulated mechanically reproduces all or part of a patient’s pain symptoms [93,94]. The hypothesis that MPS causes at least a subset of PIFP suggests an obvious direction for further study. An examination of patients with PIFP for myofascial trigger points that reproduces the patient’s pain is clearly needed. Furthermore, treatment by inactivation of such myofascial trigger points found in PIFP patients by currently known effective techniques that include treatment of neck and shoulder pain, including cervicogenic myofascial pain [95,96,97,98,99,100,101] should be studied in order to see if PIFP is diminished or eliminated by inactivation of TrPs. A study such as this would allow a determination to be made about the relevance of myofascial trigger points in PIFP. If indeed myofascial trigger points play a role in PIFP, then the next step would be to identify the causes of such trigger points and to address those underlying problems.

## 4. Conclusions

Trigeminal neuralgia and paroxysmal idiopathic facial pain are two important causes of facial pain, patient distress, and disability. Despite clear differences in their presentation, there is a gray area in which there may be an apparent overlap of symptoms. However, this is likely not to be the case, but represents confusion related to the diagnosis of PIFP. The guidelines for the diagnosis and management of classical TN are clear. However, the diagnostic steps and the management of PIFP are in evolution, as the cause of PIFP remains to be established in most cases. Myofascial pain syndrome associated with trigger points in the head and neck muscles resembles PIFP in many respects, and should be considered as a possible cause, at least in a subset of PIFP.

## Figures and Tables

**Figure 1 ijerph-17-07012-f001:**
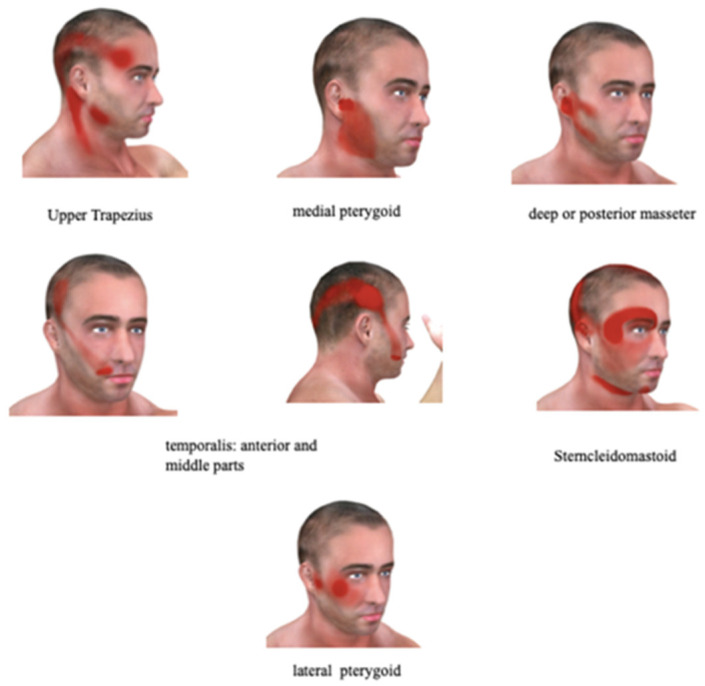
The referred pain patterns of relevant neck and head muscles that refer pain to the face in the trigeminal nerve distribution. The names of the muscles whose referred pain pattens are shown are noted under each individual illustration. Note that the sternocleidomastoid muscle can refer to the side of face primarily in the distribution of the second division of the trigeminal nerve in addition to its more common referral over the eye in the distribution of the first division. Many of these muscles are associated with the function of the temporomandibular joint and may be activated in disorders of the joint as well as in situations of bruxism and of clenching. (Images from C Triggerpoints 3D, used with permission).

**Table 1 ijerph-17-07012-t001:** The ICHD-3 criteria for the diagnosis of trigeminal neuralgia (ICHD, Cephalalgia 2013, 33,774–775).

A.	Diagnostic Criteria:
B.	Recurrent paroxysms of unilateral facial pain in the distribution of one or more ivisions of the trigeminal nerve, with no radiation, and fulling criteria B and C
C.	Pain has all of the following characteristics
	1. Occurring in one or more trigeminal nerve divisions, without radiation beyond the trigeminal distribution
	2. Paroxysms lasting from a fraction of a second to two minutes
	3. Severe intensity
	4. Electric shock like, shooting, stabbing or sharp
D.	Precipitated by innocuous stimuli within the affected trigeminal distribution
E.	No clinically evident neurologic deficit
F.	Not better accounted for by another ICHD-3 diagnosis

**Table 2 ijerph-17-07012-t002:** Summary of commonly used therapies for trigeminal neuralgia.

Modality	Assessment	Comments
**Pharmacologic Therapy**	Carbamazepine: moderate level of evidence for long-term benefit, but loss of benefit (failure rate of 50% long term) Other anticonvulsant drugs: oxcarbazepine, lamotrigine, gabapentin—commonly used but low quality or insufficient evidence re: benefit	High degree of adverse effects with carbamazepine
**Peripheral Nerve Intervention**	Percutaneous rhizotomy (glycerol): high level of evidence for long-term benefit Radiofrequency thermocoagulation: high level of evidence for long-term benefit Balloon compression: high level of evidence for long-term benefit	Loss of benefit over time for all three techniques Low incidence of serious adverse effects, but anesthesia dolorosa can be a serious adverse effect No agreement on the optimal temperature for radiofrequency thermocoagulation
**Botulinum Toxin**	High quality of evidence for benefit	Low incidence of transient side effects, but treatment must be repeated to maintain benefit
**Gamma Knife Radiosurgery**	High quality of evidence in favor of long-term benefit. Benefit falls by almost half in 5–10 years, but treatment can be repeated	Onset of improvement is delayed from 2 to 6 months after treatment Low incidence of adverse effects is increased with repeated treatment
**Microvascular Decompression**	High level of evidence for long-term improvement that is maintained over 5 years	Low incidence of adverse effects Endoscopic microvascular decompression has a higher rate of benefit and a lower rate of recurrence with fewer adverse effects than traditional open microvascular decompression

**Table 3 ijerph-17-07012-t003:** Persistent idiopathic facial pain (PFIP). The ICHD-3 criteria for the diagnosis of persistent idiopathic facial pain (ICHD, Cephalalgia 2013, 33,782).

A. Facial and/or oral pain fulling criteria B and C
B. Recurring daily for >2 h per day for >3 months
C. Pain has both of the following characteristics
1. poorly localized, and not following the distribution of a peripheral nerve
2. dull, aching or nagging quality
D. Clinical neurological examination is normal
E. A dental cause has been excluded by appropriate investigations.
F. Not better accounted for by another ICHD-3 diagnosis.

**Table 4 ijerph-17-07012-t004:** Treatments for PIFP. The following treatments are used or have been proposed for use in treating PIFP. Unfortunately, none have sufficient evidence available to make an evidenced-based recommendation for treatment.

Tricyclic Antidepressants (amitriptlyline)
Serotonin norepinephrine reuptake inhibitors
(duloxetine)
(venlefaxine)
Antiepileptics (i.e., lamotrigine)
Cannabinoids
Low-level laser
Cognitive behavioral therapy
Temporomandibular joint dysfunction and gnathic dysfunction
Sphenopalatine ganglion block

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
