# Peer review of "Chronic Facial Pain: Trigeminal Neuralgia, Persistent Idiopathic Facial Pain, and Myofascial Pain Syndrome—An Evidence-Based Narrative Review and Etiological Hypothesis"

_ijerph, 2020, doi:10.3390/ijerph17197012_

Round 1

Reviewer 1 Report

This paper discussed the difference between Trigeminal neuralgia (TN) and Persistent Idiopathic Facial Pain in causes, diagnosis, and treatment protocols. The author provided the information in detail which would be helpful for differentiating these two types of chronic facial pain. However, there are numerous revisions that needed to be done.

1) Delete the "an evidence-based narrative review and eitiological hypothesis" in the Keywords session.

2) There are many words including "-" in the middle of the text, and they should be removed.

3) Fix the typos, such as "eitiological" in the title, "rdiscussion" in the Abstract, "compex" in Line 11 on page 4.

4) Delete the extra "they" in Line 13 on page 13.

Author Response

  1. Delete... in the abstract. Thank you for the suggestion. This line is deleted. 
  2. Remove the hyphens. All the hyphens have been removed. 
  3. Correct the typos. All the typos have been corrected. 
  4. The extra 'they' has been changed to 'then. 

I appreciate the reviewer's suggestions. Thank you. 

Reviewer 2 Report

Review of the manuscript entitled: “Chronnic Facial Pain: Trigeminal Neuralgia, Persis-tent Idiopathic Facial Pain, and Myofascial Pain Syndrome; an Evidence-based Narrative Review and Eitiological Hypothesis”

This is a very interesting narrative review about facial pain and its different etiologies. The manuscript is written in a quite comprehensive way. It covers the relevant literature. The manuscript moreover reflects the author`s decades of clinical experience in this particular field. The focus of the manuscript is the presence of myofascial trigger points which seems to play a role in the etiology of some cases of persistent idiopathic facial pain (PIFP).

In my view, it is a commendable endeavour to bring myofascial trigger points to closer attention as a possible etiology of PIFP in a subset of patients. Nonetheless, the present ICHD classification does not (yet) contain any information about the possible etiological role of myofascial trigger points in PIFP.

I have some suggestions for improvement:

Abstract:

While in the course of the discussion it becomes clear, that there is still much research to do to properly define the possible etiological role of myofascial trigger points in PIFP, the abstract gives the impression that myofascial trigger points are a commonly accepted etiology of PIFP. Maybe this could be rephrased.

Introduction:

P2

“It is that confusion between PIFP and TN that prompts this consideration of the two and the interface between them that seems to have been too often rather blurred.”

This information is already given one paragraph above:

P1

“That experience, repeated many times, prompted this review.”

Maybe it is sufficient to make this statement only once.

Discussion:

P3

“The two types of non-odontological causes of facial pain that appear to be the most likely to be mistaken one for the other are trigeminal neuralgia (TN) and what used to be called atypical facial pain, but that is now called persistent idiopathic facial pain (PIFP).”

At this stage I stumbled across the term “non-odontological”. In my experience, many patients with PIFP have ineffective dental treatments in their medical history. Some of these patients report that the pain after tooth extraction even became worse, and I sometimes had the impression that these patients could have a kind of neuropathic pain caused by affection of the alveolar nerve secondary to tooth extraction. However, this is difficult to be proven and therefore these patients mostly remain diagnosed with PIFP. I understand, that at this stage (of the manuscript) in order not make things even more complicated, it is better to make a clear distinction between odontological and non-odontological causes of facial pain. Maybe this item can be discussed at a later stage in the manuscript.

P3

“3.1. Trigeminal Autonomic Cephalagias”

“3.2. Cluster Headache”

These headings make it look, as if cluster headache did not belong to the trigeminal autonomic cephalagias. In the text, it becomes clear that this is not the case.  Maybe the numeration of the headings can be changed.

P13

The structure of the subheadings in TN is different from that in PIPF. Please consider inserting subheadings such as “pharmacological treatment” ecc..

Please consider also to briefly discuss neurostimulation approaches. (i.e. Jakobs M et al Acta Neurochir 2016, for review: Maniam R et al, Curr Pain Headache Rep 2016, Ajay B. et al Pain Physician 2019). Although I am not very convinced of invasive neurostimulation as a concept for PIFP the few published results are

P13

“These authors go on to say that regional myofascial pain from pericranial, masticatory and cervical muscles may give rise to pain that clearly overlaps with the pain of PIFP. However, they they say that the referred pain patterns from these muscles may refer pain to areas that should not be seen with PIFP, such as the eye, jaw or teeth, implying that if there is pain in those areas, some other cause should be considered than myofascial pain.”

Who is meant with “these authors”? Please give a reference.

P13

“I would argue, however, that in the cases of trauma with subclincal sensory changes, the diag-nosis should be traumatic trigeminal neuralgia, and not PIFP.”

Would “trigeminal neuropathy” be more precise in this context?

Formal: Within the text, there are several hyphens at the wrong place, probably caused by mistakes in the  word processing program. Moreover, there are typing errors at several occasions (i.e. in the title: “chronnic”, “eitiological” ecc..). The manuscript should therefore undergo thorough proofreading.

Author Response

1. I have changed the wording to indicate that myofascial is suggested as a possible cause.

Head and neck muscle myofascial pain syndrome is suggested as a possible cause of PIFP, a consideration that could  open a new approaches to treatment,

2. Thank you for your suggestion. I have removed the redundant sentence. 

3. I have added an additional sentence to indicate that odontological problems can cause facial pain, but are not the focus of this paper. Elsewhere, I have also made the point that TMJ-related dysfunction can cause myofascial pain and facial pain. 

(bottom of page 2): Odontologic disorders are a known cause of facial pain but will not be addressed in this review except in passing as temporomandibular joint disorders are mentioned as a cause of myofascial dysfunction in relation to PIFP.

First paragraph page 13:

  For example, trauma to the alveolar nerve as a result of tooth extraction can result in severe, persistent, pain, but in my experience is more localized to the mouth and jaw and less to the face itself, though there may also be comorbid myofascial involvement of facial muscles.

4. Thank you for the suggestion. I have renumbered the headings to make the classifications and associations clearer. 

5. I have removed the redundant sentence. 

6. I added a sentence about peripheral neurostimulation. :

New therapies continue to be proposed and tested, as for example peripheral nerve stimulation. However, studies to date of this technique are few and with very small numbers of subjects.

I did not add more because the two papers that I found had very small number of subjects, so that this is really a very preliminary consideration. Also, I could not find the paper by Ajay, but I did find a later (2019) paper, also of a very small number of subjects. Thank you for the suggestion. 

7. I gave the reference [88] for 'these authors'. 

8. I agree that trigeminal neuropathy is more suitable and made the change. 

Thank you for your helpful suggestions. 

Reviewer 3 Report

First of all, I want to note that it has been a pleasure review your manuscript. I think this is an interesting topic for clinicians who manage this prevalent condition.

In order to improve the quality of the manuscript. After reading in depth the manuscript, I would like to make some comments and ask the author several questions about.

  • The introduction begins in a second paragraph because the first is a personal reflection. The beginning of the discussion would be included in the introduction. The introduction must be improved.
  • I would include a recent article on cervical spine myofascial therapy because it is a conclusion of the paper.
  • Did any study evaluate the quality of sleep as a variable with this pathology?.

Author Response

Thank you for your suggestions. 

  1. I retitled the headings, using Foreward for the first paragraph, and Introduction for the second paragraph. I have tightened the introduction a bit. The introduction provides the rationale for the review. The details are in the discussion. 
  2. Several of the papers on treatment address treatment of cervical (neck) myofacial pain. 
  3. Sleep was not addressed as a primary contributory factor, but only as an aspect of quality of life. Therefore, I did not address sleep specifically. However, it formed part of the multidisciplinary approach to treatment of PIFP. As there were no papers that approached sleep modification as part of therapy and tracked it separately, I did not discuss it as either a factor that contributed to pain or as a form of treatment.